# Relevance analysis of MRI sequences for automatic liver tumor segmentation

**Grzegorz Chlebus**[1,2]    grzegorz.chlebus@mevis.fraunhofer.de
**Nasreddin Abolmaali**[3]    Nasreddin.Abolmaali@klinikum-dresden.de
**Andrea Schenk**[1]    andrea.schenk@mevis.fraunhofer.de
**Hans Meine**[1,4]    meine@uni-bremen.de

[1] *Fraunhofer Institute for Digital Medicine MEVIS, Bremen, Germany*

[2] *Diagnostic Image Analysis Group, Department of Radiology and Nuclear Medicine, Radboud University Medical Center, Nijmegen, The Netherlands*

[3] *Department of Radiology, Städtisches Klinikum Dresden, Dresden, Germany*

[4] *University of Bremen, Medical Image Computing Group, Bremen, Germany*

## Abstract

Explainability of decisions made by deep neural networks is of high value as it allows for validation and improvement of models. This work proposes an approach to explain semantic segmentation networks by means of layer-wise relevance propagation. As an exemplary application, we investigate which MRI sequences are most relevant for liver tumor segmentation.

**Keywords:** explainability, deep learning, segmentation, MRI

## 1. Introduction

Algorithms employing deep neural networks achieved state-of-the-art results for many computer vision tasks (Hu et al., 2018). Thanks to millions of parameters and nonlinear behavior deep models can learn very complex input-output dependencies allowing them to surpass human expert performance (Bejnordi et al., 2017). Understanding how such models work is of big importance because it allows to verify the reasoning of the system and possibly identify model or dataset problems (Selvaraju et al., 2017; Lapuschkin et al., 2019). Several approaches for classifier explainability have been proposed: guided backpropagation (Springenberg et al., 2014), layer-wise relevance propagation (LRP) (Bach et al., 2015), and gradient-weighted class activation mapping (Selvaraju et al., 2017). These methods aim at a visualization of input regions influencing the model decision and were shown to work well for image classification and reinforcement learning models (Lapuschkin et al., 2019).

In this work, we propose a method to explain semantic segmentation models by means of LRP. We apply the proposed method to analyze the importance of input MRI sequences for the task of automatic liver tumor segmentation. Our motivation is that models requiring fewer sequences would find a broader clinical application, as not all hospitals employ the same acquisition protocols. The importance analysis would allow for an informed selection of most relevant sequences that could be used to train a segmentation model.

## 2. Materials and methods

### 2.1. Layer-wise relevance propagation

Layer-wise relevance propagation allows to obtain a pixel-wise decomposition of a model decision and is applicable for most state-of-the-art architectures for image classification and neural reinforcement learning (Bach et al., 2015). LRP uses a notion of relevance $R$ which is equal to the model output for the output neurons and can be propagated to lower layers.

#### 2.1.1. LRP FOR IMAGE CLASSIFICATION

LRP can be employed to explain classification decisions for a given class $i$ by relevance propagation from the corresponding model output $y^i$ according to:

$$y^i = R = \ldots = \sum_{d \in L^l} R_d^{(l)} = \ldots = \sum_{d \in L^1} R_d^{(1)} = \sum M^i \tag{1}$$

where $l$ refers to the layer index, $L^l$ to all neurons of layer $l$, and $R_d^{(l)}$ to a relevance of neuron $d$ in layer $l$. Typically, the relevances are propagated to the input layer ($l = 1$) yielding a relevance map $M^i$, which enables visualization of input regions influencing the model decision.

#### 2.1.2. LRP FOR SEMANTIC SEGMENTATION

In order to apply LRP to semantic segmentation models, we cast the segmentation problem as a voxel-wise classification. This means that in order to explain a decision of a segmentation model for a given output region $A$, we propose to compute the input relevance maps according to Eq. 1 for each considered output location $a \in A$. Then the relevance map $M^i$ explaining the model decision for class $i$ in the region $A$ can be calculated as:

$$M_A^i = \sum_{a \in A} \frac{M_a^i}{\sum M_a^i} \tag{2}$$

We normalize $M_a^i$ by its sum to ensure that each output location $a$ equally contributes to the final relevance map $M^i$.

### 2.2. Liver tumor segmentation model

We train and evaluate a 3D u-net (Çiçek et al., 2016) model with a 6-channel input and 2-channel output using MRI data of 69 patients (49 training, 20 evaluation) with primary liver cancer and/or liver metastates acquired on a 3T MRI scanner (GE Healthcare, USA) at Städtisches Klinikum Dresden, Germany. Imaging data of each patient contains 6 MRI sequences: T2, non contrast enhanced T1 (plain-T1), and four dynamic contrast enhanced (DCE) T1 images acquired 20 s (T1-20s), 60 s (T1-60s), 120 s (T1-120s), and 15 min (T1-15min) after contrast agent administration (Gd-EOB-DTPA). Reference tumor segmentation was performed manually by an experienced radiologist assistant on the last DCE phase. All sequences were motion corrected using a non-rigid registration using the T1-15min image as reference (Strehlow et al., 2018).

### 2.3. Relevance analysis of MRI sequences

We use Eq. 2 to compute a relevance map $M$ for both background ($i = 0$) and tumor ($i = 1$) class: $M = M_B^0 + M_T^1$. We choose regions $B$ and $T$ classified as background and tumor, respectively, to contain the same amount of voxels to prevent bias towards the more frequent background class. The importance of an input channel is corresponding to a global sum of a corresponding channel in the relevance map $M$. As absolute values of $M$ have no meaning, we normalize $M$ such that its values sum up to 1.

## 3. Results and discussion

The relevance distribution across input MRI sequences for 20 test patients is shown in Fig. 1. According to mean values, the most important sequence was T1-120s and the least important T1-20s. The biggest differences in attributed relevance were observed for T2 and T1-15min sequences. The relevance attribution for other sequences follows almost the same pattern for all test patients. The model learned to use information from all inputs as no sequence received a zero relevance. The T1-15min sequence, which was used to create reference segmentations, was not the most relevant for all test patients, which was contrary to our expectations.

## 4. Conclusion

In this work, we proposed a method to explain semantic segmentation models by means of layer-wise relevance propagation. We applied our method to analyze which MRI sequences are the most important for the task of liver tumor segmentation. Further applications of our method, for example to investigate false predictions, are future work.

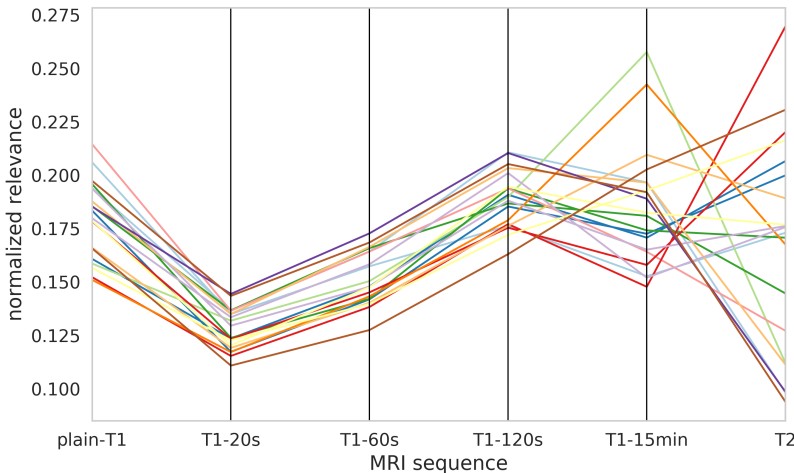

Figure 1: Normalized relevance distribution across input MRI sequences for 20 test patients denoted by different colors.

## Acknowledgments

We would like to thank Maximilian Alber, Sebatian Lapuschkin, and other authors of the iNNvestigate toolbox (Alber et al., 2018).

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
