# OpenReview forum: "Relevance analysis of MRI sequences for automatic liver tumor segmentation"
_MIDL.io/2019/Conference/Abstract — MIDL Abstract 2019_

### Official Review · AnonReviewer1 · 2019-05-01
**The application is contrived for the (otherwise interesting) methodologies under investigation.**

**Rating:** 2
**Confidence:** 2

**Review:**

The abstract reports preliminary results investigating the relevance of various MRI sequences for liver tumour segmentation, using layer-wise relevance propagation.

| Strengths:
- Explainability/interpretability of DL architectures for image or pixelwise classification (segmentation) is an interesting topic. Understanding what the network is looking at in the image and whether it has semantic meaning could provide a lot of insight into various architectures and how to improve them.

| Weaknesses:
- The focus of the work is not methodological. However, the application is a bit contrived as well in my opinion. The main advantage of such methods is in theory to provide spatial localisation of the relevant information (identify where the network is looking), but the application proposed here discards the spatial aspect.
- Ranking the relevance of MRI sequences could be achieved by much more direct methods (e.g. naively looking into the performance of various models trained using a subset of modalities) that will arguably be less challenging and more pragmatic than the "explainable AI/DL" direction.
- The validation is rather limited and yields somewhat inconclusive results.

---

### Official Review · AnonReviewer2 · 2019-05-03
**An interesting research question but with rather limited insight stemming from the abstract**

**Rating:** 3
**Confidence:** 2

**Review:**

Interpretability of DNN is an important topic, both for classification and segmentation. While most of the existing literature approaches the classification question, the authors propose a simple extension of an established method to provide some feedback on the importance of different regions and input modalities/sequences for the question of segmentation.

In more detail, the author propose to use layer-wise relevance propagation (Bach et al. 2015) to analyse the relevance of the input regions across 6 MRI sequences for the task of liver tumor segmentation. Voxel-anchored relevance maps within a given segment are averaged to achieve relevance map per class and per sequence type.

The approach of simple averaging across the region seems a bit ad hoc but might nonetheless provide some insight. Unfortunately, the authors do not highlight any result relating to spatial localisation and only focus on sequence relevance. As such, one wonders why start from an approach focusing on spatial relevance. Also, the validation of the sequence relevance is also on the low-side.

As the topic might lead to interesting discussions at the conference, I second acceptance of the abstract but this is a very weak one.

---

### Decision · Program_Chairs · 2019-05-06
**Acceptance Decision**

Accept